# Diagnosing and defining MASLD in people living with chronic hepatitis B

Emily Martyn [1,2,3], Alejandro Arenas-Pinto[3,4], Richard Gilson [3,4], Nomathemba Chandiwana[5,6], Stuart Flanagan[3,4], Douglas MacDonald[7,8], Emmanuel A. Tsochatzis[7,8], W. D. Francois Venter [9], Jennifer Manne-Goehler[10,11] & Philippa C. Matthews [1,2,3] ✉

Chronic hepatitis B (CHB) infection and metabolic dysfunction-associated steatotic liver disease (MASLD) are important contributors to the growing worldwide burden of liver disease. There is limited understanding regarding the interaction between CHB and MASLD. This is a consequence of the changing terminology for liver disease, inconsistent application of diagnostic tools, and poor understanding of global populations. In this review, we collate data on the use of diagnostic tests for identifying MASLD and associated liver inflammation or fibrosis in people living with CHB. We advocate for improved consensus on diagnosis, evidence-based monitoring and risk stratification, enhanced access to interventions and reduced health inequity.

## Epidemiology and disease burden

Chronic liver disease is a substantial contributor to global mortality, with causes that include viral hepatitis infection and metabolic liver disease[1]. Chronic hepatitis B (CHB) is responsible for an estimated 1.1 million deaths per year[2], despite the long-standing availability of a cheap, safe, robust vaccine and generic suppressive antiviral treatment. For people living with CHB, the cumulative lifetime incidence of primary hepatocellular carcinoma (HCC) is ~20% and for cirrhosis is ~40%, but with substantial heterogeneity[3,4].

The contribution of metabolic dysfunction-associated liver disease (MASLD) to morbidity and mortality is also increasing worldwide, with over one billion people now living with overweight and obesity, and >800 million with diabetes[5,6]. As the impact of metabolic disease on liver outcomes is emerging, it is estimated that 3% of people living with MASLD will develop major liver-related complications over 20 years[7]. MASLD is also associated with ~1.5-fold increased risk of fatal and non-fatal cardiovascular events, 3-fold higher risk of developing chronic kidney disease, and increased risk of extrahepatic cancers[5]. Importantly, the prevalence of metabolic conditions is climbing in regions with high CHB endemicity, for example, in African, eastern Mediterranean, and western Pacific regions[8]. This critical nexus between CHB and MASLD will undoubtedly alter the epidemiology and morbidity of these conditions worldwide (Fig. 1).

## Terminology and definitions

The term MASLD was introduced in 2023[9] to replace potentially pejorative or stigmatising language in previous definitions, which included reference to 'alcoholic' and 'fatty' liver disease. Diagnosis of MASLD requires identification of steatotic liver disease together with at least one cardiometabolic factor (hypertension, diabetes, dyslipidaemia, or obesity). Relevant thresholds for the use of cardiometabolic factors in the diagnosis of MASLD were agreed via a Delphi consensus process, largely based on the existing definition of metabolic syndrome[9,10] (Fig. 2). This standardisation of terminology simplifies approaches to diagnosis and management and improves consistency of data collection as new therapeutics emerge. For consistency, we use the term MASLD throughout this review, even when referencing literature published prior to 2023, which uses older terminology (in contrast, in the Supplementary Tables 1−5, we use the original terms used by each article). In parallel with shifts in the MASLD field, the CHB landscape is changing, as there has been a move away from complicated classification systems and definitions, aiming to develop simplified practical approaches to diagnosis and risk-assessment, which will support more equitable access to interventions and treatment[2]. CHB refers to hepatitis B virus (HBV) infection, which has persisted for longer than six months, based on the presence of HBV surface antigen, HBsAg, and/or HBV DNA in peripheral blood. The term "person/people living with HBV" (PLWHB) is used where possible to emphasise person-centred language. Clearance of

[1]The Francis Crick Institute, London, UK. [2]Division of Infection and Immunity, University College London, London, UK. [3]Central North West London NHS Foundation Trust, London, UK. [4]Institute for Global Health, University College London, London, UK. [5]Desmond Tutu HIV Centre, University of Cape Town, Cape Town, South Africa. [6]Desmond Tutu Health Foundation, Cape Town, South Africa. [7]Sheila Sherlock Liver Centre, Royal Free Hospital NHS Foundation Trust, London, UK. [8]Institute for Liver and Digestive Health, University College London, London, UK. [9]Ezintsha, Faculty of Health Sciences, University of the Witwatersrand, Johannesburg, South Africa. [10]Division of Infectious Diseases, Brigham and Women's Hospital, Harvard Medical School, Boston, MA, USA. [11]Medical Practice Evaluation Center, Massachusetts General Hospital, Harvard Medical School, Boston, MA, USA. ✉e-mail: philippa.matthews@crick.ac.uk

**Fig. 1 | Chronic Hepatitis B (CHB) and metabolic dysfunction-associated steatotic liver disease (MASLD) are major global health challenges.** Figures based on 2019 estimates from previously published reviews[5,102–106]. Created in BioRender. "Martyn, E. (2026) https://BioRender.com/wvagdlp".

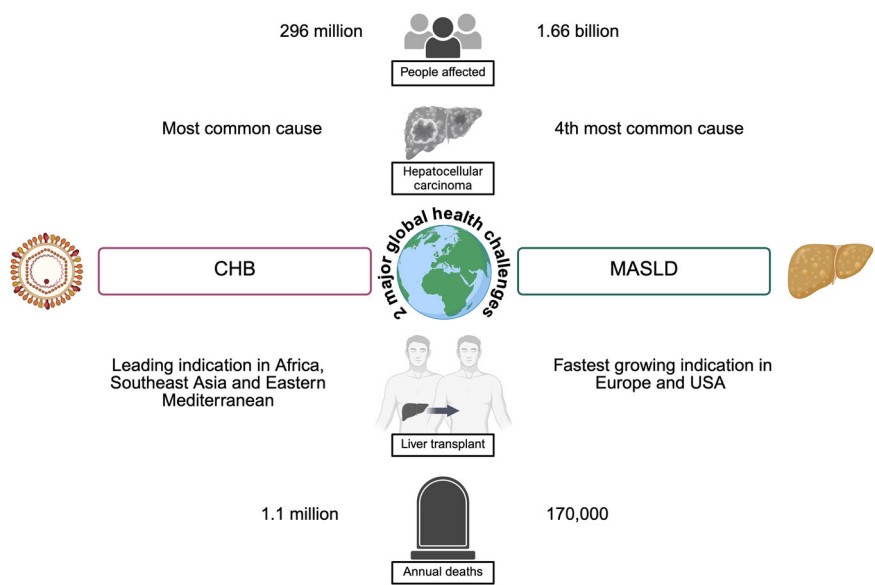

hepatitis B surface antigen (HBsAg) in the context of CHB is termed 'functional cure'.

## Challenges for the field

To date, algorithms for the classification of hepatic steatosis may lead to inconsistencies, due to high population heterogeneity and complex overlapping risk factors for the evolution of liver disease, which include genetics, environment, diet, socioeconomic factors, and the influence of migration on risk[11]. It is also notable that the evidence used to establish thresholds is not representative of all populations, with different tools used according to location and resources, and scarce data from World Health Organization (WHO)Africa and Eastern Mediterranean regions (where MASLD and CHB are significant public health threats) (Supplementary Table 1). Insufficient data from resource-constrained settings reflecting key populations contributes to gaps in our understanding of epidemiology, clinical impact, and public health implications of the MASLD/CHB interplay[12,13]. Thus, the field is complicated by heterogeneous (and in some cases, conflicting) data. Despite increasing evidence informing diagnosis, treatment, and public health interventions for both CHB and MASLD, profound health inequities persist.

## Aims of the review

In this narrative review, we consider the diagnosis and risk assessment of MASLD in PLWHB. Our aim is to summarise existing tools and evidence, and to highlight the need for concordance in the use and interpretation of diagnostic tests, with special emphasis on those that can be made accessible across diverse settings to facilitate better assessment and provision of evidence-based healthcare. By unifying classification and improving understanding of the intersection of these two important conditions, we can advance an understanding of the biology, clinical implications and treatment pathways, tackle health inequities, and identify areas for specific action.

## Evidence for diverse outcomes in PLWHB and MASLD

To date, the impact of MASLD on liver disease and viral outcomes in PLWHB has been explored in 12 systematic reviews[14–24], which collectively incorporate data from 178 studies (Table 1). Despite the increasing body of data, conflicting evidence associates MASLD with either increased or decreased risk of the development of chronic liver disease, cirrhosis, HCC, and poor treatment responses in PLWHB (Table 1, Fig. 3). The reasons behind these conflicting findings are likely to be multifactorial. One possible contributing factor is the heterogeneity of studies included in the meta-analyses. For example, older definitions of metabolic or fatty liver disease

(Fig. 2), different methods of MASLD diagnosis (e.g., imaging, biopsy, non-invasive serum-based scores), different populations, and varied study designs (e.g., cohort vs. cross-sectional) (Table 1). Another possible explanation is that risk factors have a differential impact on outcomes in CHB, with cardiometabolic factors having a dose-dependent adverse effect, while steatotic liver disease may have a protective effect[25–27]. The challenge of accurately estimating the individual causal effect estimates of these interrelated factors using routinely collected clinical data likely contributes to current ambiguity in the literature.

One consistent (albeit perhaps unexpected) observation is that functional cure occurs more frequently in the presence of MASLD[17,19]. Hypothesised mechanisms for this increased chance of HBsAg loss include apoptosis of steatotic HBV-infected hepatocytes, fatty acid inhibition of HBsAg secretion, and increased MASLD-related inflammation boosting immune-related HBsAg seroclearance[28–31]. However, these mechanisms are speculative, and more robust experimental evidence is required.

Outcomes of CHB are known to be influenced by individual attributes (age at infection, sex, genetics), infection attributes (HBV genotype, e-Antigen status, viral load), comorbidity including coinfections (HIV, hepatitis D virus), and antiviral treatment, all of which are reflected in clinical practice guidelines[2,32,33]. New nomenclature recognises the need to identify liver inflammation in the presence of MASLD (termed metabolic dysfunction-associated steatohepatitis, 'MASH'; Fig. 2)[34,35], irrespective of the presence or absence of CHB, although it is the severity of fibrosis rather than inflammation that is associated with long-term outcomes[36]. On these grounds, there is particular focus on identifying MASH with fibrosis ('at-risk MASH'), to inform earlier intervention or enhanced monitoring, and to identify a population of interest for clinical trials.

## Assessment of MASLD in PLWHB

The conventional gold standard for diagnosis of steatotic liver disease (SLD) and fibrosis/cirrhosis is histopathological examination of a liver biopsy (Supplementary Table 2), with steatotic liver disease defined by >5% of hepatocytes containing fat deposition, and graded as S1, S2 or S3 based on fat deposition in 5–33%, 34–66%, and >66% of hepatocytes, respectively[37]. However, a biopsy samples only a tiny fragment of liver tissue, requires expert operators, is costly, and associated with rare but important complications, which means it should be undertaken in a specialist centre with clinical expertise on site.

To reduce risk and improve access, there is a move towards non-invasive tests (NITs), which can be made safely and universally accessible[12,38]. NIT results can be incorporated into algorithms to identify individuals at the highest risk of complications (including identifying

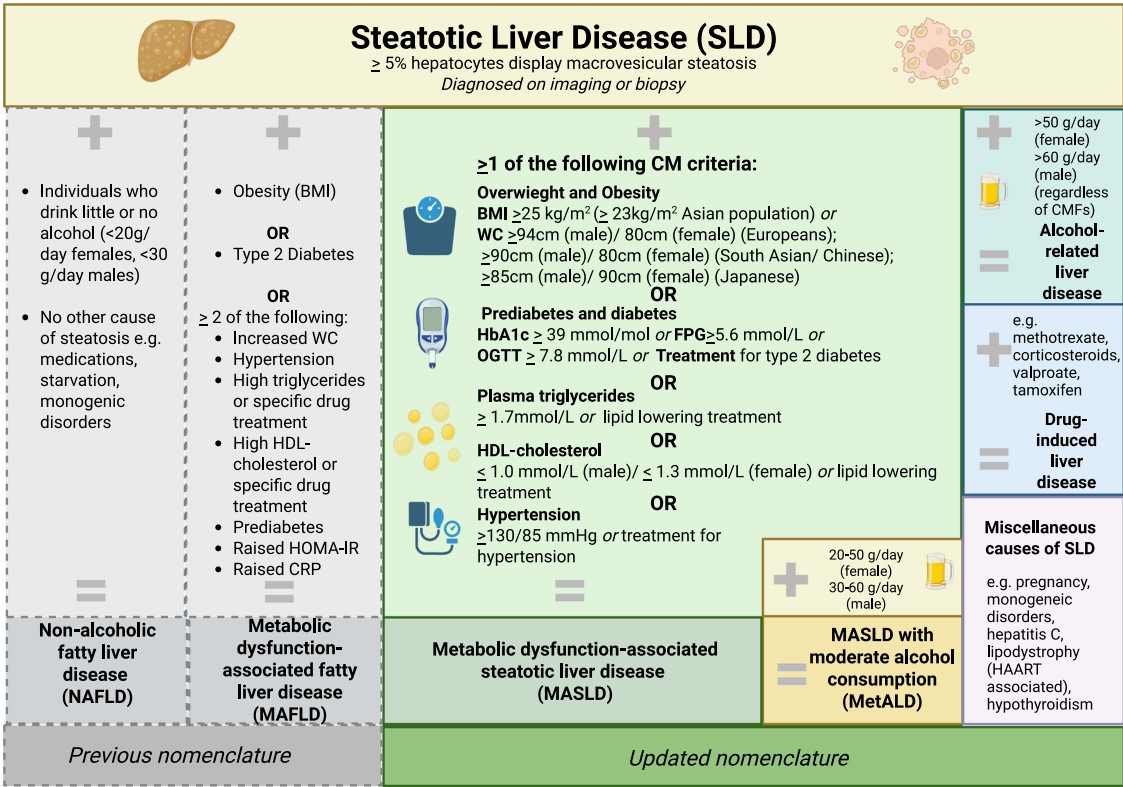

**Fig. 2 | Schematic to summarise definitions of steatotic liver disease and cardiometabolic factors.** Metabolic dysfunction Associated Steatotic Liver Disease (MASLD) replaces prior terminology of 'metabolic dysfunction-associated fatty liver disease' (MAFLD) and 'non-alcoholic fatty liver disease' (NAFLD)[9]. Although definitions have changed, there is high concordance between MASLD and NAFLD[107]. Metabolic dysfunction-associated steatohepatitis ('MASH') is defined as MASLD with liver inflammation and/or fibrosis. Created in BioRender. Martyn, E. (2026) https://BioRender.com/yv9c4yy. BMI body mass index, CRP C-reactive protein, FPG fasting plasma glucose, HAART highly active antiretroviral therapy, HbA1c haemoglobin A1c, HDL high density lipoprotein, HBV hepatitis B virus, HOMA-IR homeostasis model assessment of insulin resistance score, MetALD metabolic dysfunction-associated steatotic liver disease with moderate alcohol consumption, OGTT oral glucose tolerance test, WC waist circumference. MAFLD cut-offs: BMI ≥ 25 kg/m² (WC ≥ 25 kg/m² Asian population); Type 2 diabetes diagnosed "according to widely accepted international criteria"; WC Caucasian: ≥102 cm (male)/88 cm (female), Asian: ≥90 cm (male)/80 cm (female); BP ≥ 130/85 or treatment for hypertension; Prediabetes HbA1c 39–47 mmol/mol; Triglycerides ≥1.7 mmol/L; HDL-cholesterol ≤ 1.0 mmol/L (male), ≤1.3 mmol/L (female); HOMA-IR ≥ 2.5; CRP > 2 mg/L[9,47,108].

selected individuals who may benefit from biopsy, in situations where this can be accessed). However, the coexistence of CHB and MASLD may affect NIT performance, and many MASLD NITs were developed and validated in studies that excluded CHB.

## Imaging modalities that detect fibrosis and steatosis

Ultrasound and magnetic resonance imaging (MRI) are the two main modalities used to assess liver steatosis and fibrosis (Table 2). To assess steatosis, the echogenicity of the liver and renal parenchyma can be compared, with increased hepatic echogenicity corresponding to steatosis. Quantitative scores can be derived based on attenuation of the ultrasound beam as it passes through liver tissue, for example, by controlled attenuation parameter (CAP), which was first validated in humans in 2010[39]. The most widely used ultrasound technology for liver stiffness measurement uses vibration-controlled transient elastography (VCTE). To measure VCTE, a mechanical pulse is transmitted through the liver, producing a shear wave that is converted into a quantitative value[40], which is used to quantify and grade fibrosis. This shear wave is also influenced by the presence of inflammation. Both CAP and VCTE are most widely available through hardware produced by Fibroscan™ (Echosens, France), although new tools are coming onto the market.

CAP/VCTE measurements of steatosis and liver stiffness have moderate-good accuracy compared to biopsy, and can be collected as quick, mobile, point-of-care tests by a wide range of personnel, as these measures require simple training and are not limited to expert healthcare workers. Despite these advantages, the global community is far from achieving

equitable access, due to the high costs associated with procurement and maintenance of equipment, running expenses for annual recalibration, equipment upgrades, insurance, and employment and training of staff. Other limitations include reduced accuracy caused by factors which affect the elastic properties of the liver (e.g., transaminitis, congestive heart failure, post-prandial hyperaemia, cholestasis) and impede the transmission of shear waves (e.g., ascites)[41]. The standard M probe may not perform well in the context of overweight/obesity; therefore, Fibroscan™ introduced the XL probe for use when the skin-to-capsule distance is >25 mm[42].

Magnetic resonance spectroscopy - proton density fat fraction (MRS-PDFF) measures the different resonance frequencies between water and fat proton signals to calculate the proton density fat fraction, defined as the fraction of mobile protons in the liver attributable to fat[43]. MRS-PDFF is regarded as the non-invasive gold standard tool for fat quantification, but requires intensive data post-processing[43,44]. Magnetic resonance imaging - proton density fat fraction (MRI-PDFF) has been developed over the last 15 years and is based on the same underlying physics concepts as MRS-PDFF, but is more practical to measure in a clinical setting[43]. Magnetic resonance elastography (MRE) provides the benefit of assessing fibrosis across the whole liver[45]. To date, there is no evidence for the use of MRS/MRI-PDFF or MRE specifically for diagnosis of MASLD in PLWHB, and since this approach is not widely available or affordable, it remains in the research domain and is unlikely to move to routine clinical practice in the short-to-medium term.

Clinical and public health applications of CAP are currently limited by lack of consistency for CAP thresholds in comparison to assessment by

**Table 1 | Summary of 12 meta-analyses of steatotic liver disease (SLD) in people living with chronic hepatitis B (PLWHB) infection**

| First author; date of publication | Number of studies | Study country (number of studies by country) | SLD diagnosis method (number of studies by method) | SLD prevalence |
|---|---|---|---|---|
| Machado; 2011[14] | 17 | Australia (1), China (4), Greece (4), India (2), Iran (1), Korea (1), Romania (1), Turkey (2), USA (1) | Biopsy (17) | 29.6% (range 14–70%) |
| Jiang; 2021[15] | 98 | Brazil (2), China (60), France (1), Greece (1), Hong Kong (1), India (1), Iran (3), Israel (1), Italy (2), Korea (5), Malaysia (1), Pakistan (2), Philippines (1), Poland (1), Romania (2), Singapore (1), Spain (1), Thailand (1), Tunisia (1), Turkey (6), USA (3) | CT (1); CAP (13); Biopsy (62); MR (1); Mix (3), US (17) | 34.9% (95% CI 32.0–37.9) |
| Zheng; 2021[16] | 49 | Brazil (1), China (23), Greece (2) Hong Kong (2) India (1), Iran (1), Italy (1), Korea (4), Romania (2), Spain (2), Taiwan (2), Thailand (2), Turkey (6), USA (2) | Biopsy (31); US (10); CAP (4); Mix (4) | 32.8% (95% CI 28.9–37.0) |
| Mao; 2023[17] | 34 | Canada/ Netherlands (1), China (13), Greece (1), Hong Kong (2), Israel (1), Korea (8), Malaysia (1), Singapore (1), Taiwan (2), Thailand (1), Tunisia (1), Turkey (1), USA (1) | FLI (1), CAP (5); Biopsy (22); US (5); Mix (1) | N/A |
| Zhou; 2023[18] | 85 | China (59), Hong Kong (2), India (2), Iran (3), Israel (1), Korea (5), Malaysia (1), Pakistan (2), Taiwan (2), Thailand (1), Turkey (7) | CAP (16); Biopsy (51); Mix (1); MR (1); US (15) | 36.5% (95% CI 33.7–39.3) |
| Wong; 2023[19] | 16 | Canada/Netherlands (1), China (4), Hong Kong (1), Israel (1), Korea (3), Singapore (1), Taiwan (3), Turkey (1), Australia) | CAP (4); Biopsy (7); US (4); Mix (1) | N/A |
| Han; 2023[22] | 11 | Canada/Netherlands (1), China (1), China/USA (1), Hong Kong (1), Korea (4), Singapore (1), Taiwan (2) | FLI (1); CAP (1); Biopsy (4); US (2); Mix (2); Not recorded (1) | N/A |
| Shen; 2024[21] | 18 | Canada (1), Canada/Netherlands (1), China (3), Hong Kong (1), Israel (1), Korea (5), Singapore (1), Taiwan (4), Thailand (1) | FLI/HSI/SIHBV (1); CAP (3); Biopsy (8); Mix (3); US (3) | N/A |
| Rui; 2024[20] | 10 | China (5), Korea (1), Thailand (1), Turkey (2), USA (1) | CAP (2); Biopsy (4); US (3); Mix (US, CT, MR) | 42.2% (range: 33.7–66.7%) |
| Liu; 2024[23] | 11 | China (6), France (1), Korea (3), USA (1) | CAP (1); Biopsy (6); US (3); Mix (US, CT, MR, Biopsy) (1) | 37% |
| Zeng; 2024[91] | 11 | China (6), Korea (1), Taiwan (1), USA (1), Turkey (1), global (1) | CAP (2); Biopsy (5); US (2), CT Liver/ Spleen Ratio (1), Mix (1) | N/A |
| Zhu; 2025[24] | 24 | China (15), Canada (3), Taiwan (2), Korea (2), Singapore (1), Israel (1) | Biopsy (16), US (6), FLI (1), N/A (1) | N/A |

Outcomes are summarised in Fig. 3.
CAP controlled attenuation parameter, CI confidence interval, CT computer tomography, FLI fatty liver index, HSI hepatic steatosis index, Mix mixed diagnostic modalities, MR magnetic resonance, N/A not available, SIHBV steatosis index in people living with HBV, SLD steatotic liver disease, US ultrasound, USA United States of America.

**Fig. 3 | Summary of the results of the systematic reviews and meta-analyses investigating the impact of metabolic dysfunction-associated steatotic liver disease (MASLD) on liver outcomes in chronic hepatitis B (CHB).** -- No impact on outcome, ↑ outcome increased in people living with CHB and MASLD compared to CHB alone, ↓ outcome decreased in people living with CHB and MASLD compared to CHB alone. The first author of the meta-analysis and the year published are stated. A list of these meta-analyses is presented linked to full citations in Table 1.

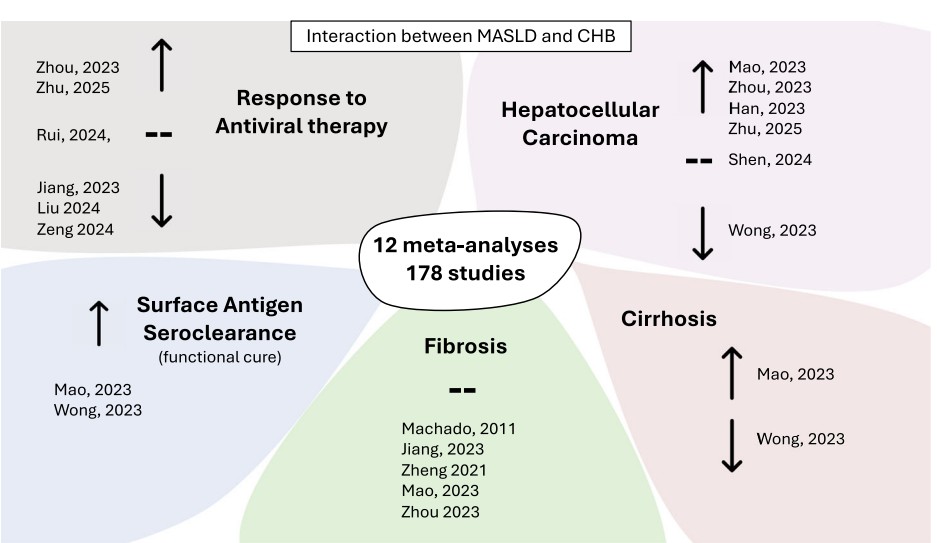

### Table 2 | Imaging-based tests used to diagnose and assess metabolic dysfunction-associated steatotic liver disease (MASLD) in people living with chronic hepatitis B (PLWHB)

| Test | Advantages | Disadvantages | Data for CHB + MASLD |
|---|---|---|---|
| Ultrasound (US) | - Widely available in high-income settings. <br> - Ability to identify multiple pathologies (SLD, fibrosis, cirrhosis, HCC). | -Operator-dependent. <br> -Only detects advanced fibrosis. <br> - Reduced sensitivity for steatosis affecting <20% hepatocytes[92] <br> - Less accurate in obesity. <br> -Expensive equipment and training required, limiting use in low-income settings. | Only one small study identified specifically for diagnosing SLD in CHB (Total 109 PLWHB, 48 of whom had SLD). US had 89% sensitivity/ 94% specificity to detect severe SLD. This is in line with a large meta-analysis detected SLD in people with mixed liver disease aetiology[93,94]. |
| Quantitative US (e.g. measured by Fibroscan™) | - Point-of-care test, portable, minimal training. <br> - Ability to measure fibrosis (VCTE) and steatosis (CAP). <br> - More sensitive than ultrasound for SLD and fibrosis measurement. <br> -Different probes available for people with overweight/obesity. | - No consensus diagnostic thresholds. <br> - Measurements may be influenced by disease aetiology e.g., steatosis. <br> -Factors which influence elastic properties of the liver (e.g., ascites, transaminitis, cholestasis, hepatic congestion, and post-prandial hepatic hyperaemia) or affect shear wave transmission (e.g. ascites, obesity, small rib spaces) can impact measurement accuracy. <br> - Expensive, limiting global implementation. | CAP (steatosis): <br> - 2 meta-analyses (total 1258 PLWHB) – different diagnostic thresholds[49,52] <br> VCTE (fibrosis): <br> - Fibroscan is widely validated for CHB + MASLD. <br> - Some evidence that liver stiffness is overestimated in severe steatosis, consensus thresholds not agreed upon for MASLD + CHB[55,95]. <br> - Other quantitative US methods of measuring liver stiffness, e.g., 2D-shear wave elastography, are less well validated[96]. <br> -VCTE and CAP have superior diagnostic performance to serum-based NITs[53,59,97]. |
| Magnetic resonance spectroscopy - proton density fat fraction (MRS-PDFF)/ magnetic resonance imaging - proton density fat fraction (MRI-PDFF/)/magnetic resonance elastography MRE | - MRS-PDFF: gold standard MR measurement of hepatic triglyceride content[43]. <br> -MRI-PDFF: more widely available than MRS-PDFF, an accurate measure of SLD across the whole liver[43]. <br> - MRE gives an accurate measure of fibrosis[98]. | - MRS-PDFF: research technique only[43]. <br> - All MR techniques have limited availability due to high cost and expertise required to perform and interpret scan. <br> -Claustrophobia and metallic implanted devices are contraindications. | No data specifically for MASLD + CHB. |

*CAP* controlled attenuation parameter, *CHB* Chronic hepatitis B, *HCC* hepatocellular carcinoma, *MRE* magnetic resonance elastography, *MRI-PDFF* magnetic resonance imaging - proton density fat fraction, *MRS-PDFF* magnetic resonance spectroscopy - proton density fat fraction, *NIT* non-invasive test, *SLD* steatotic liver disease, *US* ultrasound, *VCTE* vibration-controlled transient elastogrpahy.

biopsy (Fig. 4). European MASLD guidelines suggest distinct thresholds for distinguishing between steatosis grades (with cut-offs to distinguish S0/1, S1/2 and S2/3 thresholds), based on individual patient data meta-analysis including almost 4000 individuals with multi-aetiology liver disease, including >1000 PLWHB. In contrast, American guidelines advocate for a single rule-in CAP threshold, which sits in the European S3 category (based on a single-centre study with 119 participants, excluding PLWHB)[46–49]. The decision to select a single binary threshold recognises the limited accuracy of CAP to distinguish between steatosis grades, in particular S2–S3[50,51].

Neither European nor American guidelines acknowledge the potential influence of CHB on CAP thresholds, although meta-analysis suggests that CHB independently lowers CAP scores when compared to other factors that influence liver pathology: −10 dB/M (95% confidence interval −17 to −3.6) compared to steatosis alone and −17.2 dB/M (95% CI −34.8 to 0.2) compared to alcohol-related liver disease[49,52]. When considering viral hepatitis alone (either CHB or chronic hepatitis C (CHC)), optimal CAP thresholds were lower than in studies considering mixed liver disease or MASLD alone (Fig. 4; Table 2; Supplementary Table 3).

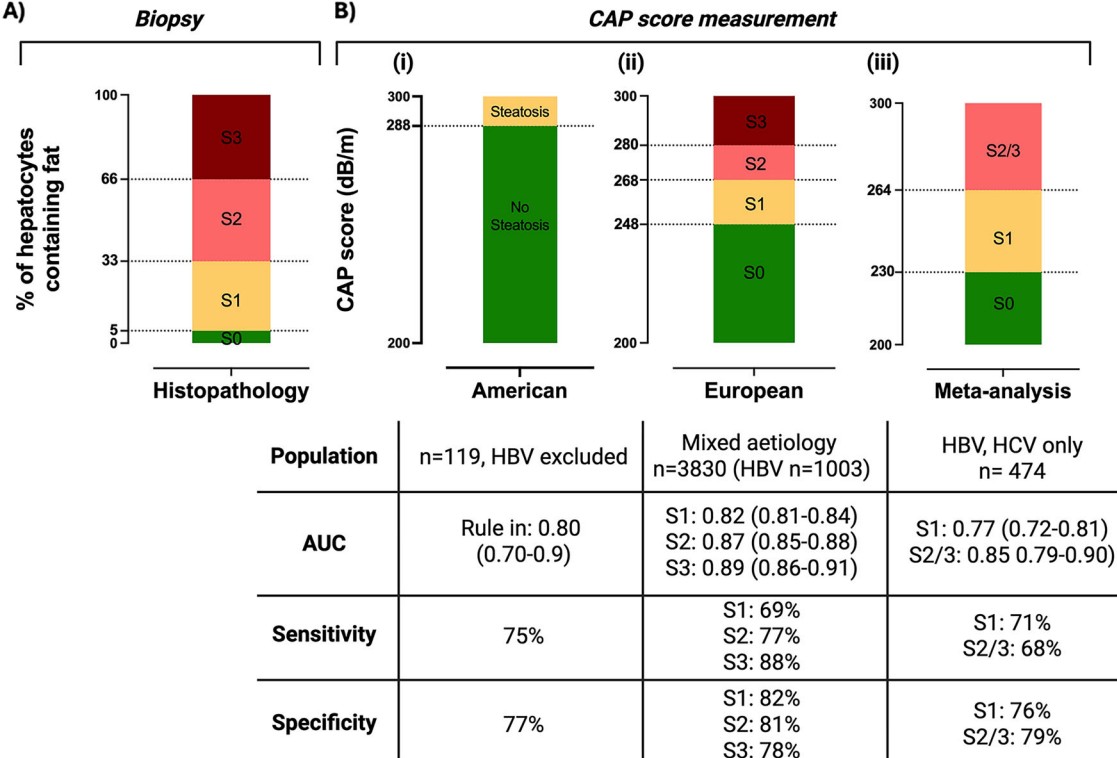

| | | | |
|---|---|---|---|
| **Population** | n=119, HBV excluded | Mixed aetiology n=3830 (HBV n=1003) | HBV, HCV only n= 474 |
| **AUC** | Rule in: 0.80 (0.70-0.9) | S1: 0.82 (0.81-0.84) S2: 0.87 (0.85-0.88) S3: 0.89 (0.86-0.91) | S1: 0.77 (0.72-0.81) S2/3: 0.85 0.79-0.90) |
| **Sensitivity** | 75% | S1: 69% S2: 77% S3: 88% | S1: 71% S2/3: 68% |
| **Specificity** | 77% | S1: 82% S2: 81% S3: 78% | S1: 76% S2/3: 79% |

**Fig. 4 | Illustration of thresholds applied for defining and grading liver steatosis.** Criteria are shown using **A** histopathology and **B** non-invasive assessment by controlled attenuation parameter (CAP) measurement, based on (i) American guidelines (based on a single centre study with 119 patients, excluding people living with chronic hepatitis B (PLWHB)[47,49], (ii) European guidelines (based on meta-analysis data including multi-aetiology liver disease, including over 1000 PLWHB[52]), and (iii) by meta-analysis data with viral-hepatitis specific thresholds[49]. Colours indicate grades of steatosis: Green—S0, Yellow—S1, Light red—S2, Dark red—S3. AUC area under the receiver operator curve, CAP controlled attenuation parameter, CHB chronic hepatitis B, CHC chronic hepatitis C, S1 mild hepatic steatosis (consistent with fat deposition in 5–33% of hepatocytes); S2—moderate hepatic steatosis (fat deposition in 34–66% of hepatocytes), S3—severe steatosis (fat deposition in >66% of hepatocytes).

A systematic review and meta-analysis conducted by the WHO led to a recommendation of elastography scores >7 kPa to detect liver fibrosis (≥F2) and >12.5 kPa to detect cirrhosis (F4) in CHB[2]. Some studies of MASLD in the context of CHB report that there is a higher elastography threshold to diagnose advanced fibrosis in people with CAP ≥ 268 dB/m compared to those with lower CAP scores (<268 dB/m) (8.8 vs. 7.0 kPa, respectively)[53–57]. This remains a topic of debate; while studies reaching this conclusion used only M probes to measure CAP, and excluded obesity, the use of the correct probe size may influence conclusions, and other studies have found no such relationship between CAP score and fibrosis threshold[58].

While data to suggest variation in CAP and VCTE values according to disease threshold may create confusion, especially as there are global efforts to simplify CHB treatment guidelines, it is not possible to suggest a single cutoff threshold based on current data. Further research is needed to understand optimal thresholds for use in clinical practice.

### Clinical and laboratory-based scores to identify liver disease

Simple scoring systems are based on routine laboratory parameters, typically using alanine transaminase (ALT), aspartate aminotransferase (AST), and platelet counts, with or without other demographic or biometric data. These approaches provide an assessment of liver health, aiming to identify and risk-stratify steatosis and/or fibrosis. However, there is variable data for the influence of CHB on the performance of these scores.

Steatosis scores are based on laboratory tests, anthropometric measurements, and cardiometabolic risk factors, e.g., NAFLD-liver fat score (NAFLD-LFS), fatty liver index (FLI), hepatic steatosis index (HSI), and others (summarised in Table 3; Supplementary Table 3). Most of these approaches have not been assessed in the context of CHB, which has been

justified because CHB infection is not steatogenic. However, some validation has been undertaken in small studies of CHB[59,60], and one score has been specifically proposed for use in the setting of CHB, the 'steatosis index in patients with HBV' (SIHBV). SIHBV was derived from a cohort of 182 Chinese individuals evaluated with liver biopsy, among whom 59 had >5% steatosis histologically[60]. In a validation cohort that incorporated liver biopsies, SIHBV outperformed other NITs and had a better sensitivity for diagnosing mild steatosis (<20%) than ultrasound. This approach was published in 2016, but has yet to be validated in larger cohorts, presumably reflecting the practical difficulty, cost, and ethical implications of undertaking a biopsy.

Fibrosis scores are summarised in Table 4. AST to platelet ratio index (APRI) > 0.5 is endorsed by WHO CHB treatment guidelines for the identification of individuals with ≥F2 fibrosis[2], and Fibrosis-4 (Fib4) is recommended for use by primary care in European MASLD guidelines to identify F3 fibrosis[47]; in the presence of cardiometabolic risk factors, individuals with Fib4 > 1.3 require further non-invasive fibrosis testing while those with >2.67 may benefit from specialist hepatology assessment[47]. Enhanced liver fibrosis (ELF) is endorsed as an alternative to VCTE to assess for fibrosis in MASLD, and several studies have demonstrated good diagnostic performance in CHB; however, no studies have investigated accuracy for individuals living with concomitant MASLD/CHB[47,61].

In studies that have compared serum-based fibrosis scores in PLWHB and MASLD, their performance is generally equivalent, even when compared across different metabolic liver disease definitions (NAFLD, MAFLD, and MASLD)[53,62–64]. One exception was in MASLD with moderate alcohol intake (MetALD), where gamma-glutamyl transpeptidase (GGT) to platelet ratio (GPR) and APRI had superior performance to Fib4 and VCTE[64]. In

**Table 3 | Serum-based non-invasive tests to diagnose metabolic dysfunction-associated steatotic liver disease (MASLD) in people living with chronic hepatitis B (CHB)**

| Test | Components | Data for CHB + MASLD | Comments |
|---|---|---|---|
| Hepatic steatosis index (HSI) | ALT, AST, BMI, sex, diabetes | 4 studies (total $n = 1360$)[59,60,99,100]. AUC range 0.63–0.79 (reference: liver biopsy), sensitivity 60–100%/specificity 58–66%. | Externally validated. Reference test is US (less accurate than liver biopsy). |
| Fatty liver index (FLI) | GGT, triglycerides, BMI, waist circumference | 1 study ($n = 364$)[60]. AUC 0.75 (sensitivity/specificity not reported). | Externally validated. |
| Non-alcoholic liver disease liver fat score (NAFLD-LFS) | ALT, AST, fasting insulin level, presence of T2DM, and metabolic syndrome | 1 study ($n = 302$)[99]. AUC 0.70, Sensitivity 88%, specificity 52%. Similar to TyG, HSI, and VAI in a head-to-head comparison. | Externally validated. Reference test is MRI-PDFF (arguably, the best test for diagnosing SLD). Fasting insulin is inconvenient to measure. |
| Visceral adiposity index (VAI) | Triglycerides, high-density lipoprotein, BMI, WC | 1 study ($n = 302$)[99]. AUC 0.86 (95% CI 0.56–0.79); sensitivity 73%; specificity 62%. | Not widely validated. Developed as a surrogate marker of visceral adiposity. |
| Triglyceride glucose index (TyG) | Triglycerides, glucose | 1 study ($n = 302$)[99]. AUC 0.68 (95% CI 0.57–0.80); sensitivity 72%; specificity 65%. | Not widely validated. Developed as a surrogate marker of insulin resistance. |
| Fatty liver disease index (FLD) | ALT, AST, triglycerides, hyperglycaemia, BMI | 1 study ($n = 364$)[60]. AUC 0.77 (sensitivity/specificity not reported). | Not widely validated. Fasting glucose is inconvenient to measure. |
| Korean score | ALT, AST, GGT, triglycerides, BMI | 1 study ($n = 364$)[60]. AUC 0.65 (sensitivity/specificity not reported), lower than other NITs in the same population (HSI, FLI, SIHBV, FLD, LAP). | Not widely validated. |
| Lipid accumulation product (LAP) | Triglycerides, waist circumference | 1 study ($n = 364$)[60]. AUC = 0.68 (sensitivity/specificity not reported, lower than other NITs in the same population (HSI, FLI, SIHBV, FLD). | Not widely validated. Developed as an index of cardiometabolic risk, intended as a screening rather than a diagnostic tool. |
| Steatosis index in HBV (SIHBV) | Triglycerides, haemoglobin, serum uric acid, age | 1 study ($n = 364$)[60]. AUC 0.86 (0.79–0.92), higher than FLD, FLI, Korean score and LAP. Using a rule in score ≥ 0.48, 93% specificity; rule out < 0.18, 91% sensitivity. | Not externally validated. Specifically designed to detect SLD in PLWHB. Better sensitivity to detect mild steatosis (<20%) than US. |

*ALT* alanine transaminase, *AST* aspartate aminotransferase, *AUC* area under the receiver operator curve analysis, *BMI* body mass index, *CHB* chronic hepatitis B, *FLD* fatty liver disease index, *FLI* fatty liver index, *GGT* gamma-glutamyl transferase, *HBV* hepatitis B virus, *HSI* hepatic steatosis index, *LAP* lipid accumulation product, *MASLD* metabolic dysfunction-associated steatotic liver disease, *MRI-PDFF* magnetic resonance imaging - proton density fat fraction, *NAFLD-LFS* non-alcoholic fatty liver disease liver fat score, *PLWHB* people living with chronic hepatitis B, *SIHBV* steatosis index in HBV, *SLD* steatotic liver disease, *T2DM* type 2 diabetes mellitus, *US* ultrasound, *WC* waist circumference.

some datasets, GPR may perform better than other laboratory-based approaches in identifying advanced fibrosis (F3–F4)[65]. In a small Chinese study, GPR had a significantly higher AUC and negative predictive value to detect ≥F3 fibrosis compared to Fib-4 and APRI (Table 4, Supplementary Table 4)[66]. In some African cohorts, GPR has also outperformed other scores[67,68]. However, GPR remains understudied compared to other NITs and has poor performance in the setting of HBV/HIV coinfection, limiting widespread implementation[69].

Approaches to combine appraisal of steatosis and fibrosis are required for diagnosis and evaluation of at-risk MASH (Supplementary Table 5). AGILE 3+ and AGILE 4 combine VCTE with sex, diabetes, ALT, AST, platelets ± age to detect advanced fibrosis and cirrhosis, respectively, in the context of suspected MASLD[45]. Real-world European studies find that AGILE 3+ and 4 scores have similar AUCs, but classify more people living with MASLD correctly (fewer indeterminate results) than VCTE alone, but this was not replicated in PLWHB[70]. However, in a Chinese cohort of PLWHB and MASLD, AGILE 3+ demonstrated better ability to correctly classify fibrosis (fewer indeterminates and higher proportion with advanced fibrosis ruled out) than VCTE alone[63]. The NAFLD fibrosis score (NFS), which aims to identify ≥F3 fibrosis, has been validated in multiple external cohorts[71,72]. The PPDHG Score was developed to detect fibrosis specifically in antiviral naïve PLWHB and MASLD to detect advanced fibrosis (≥F3). It had a higher AUC, sensitivity, and specificity than APRI, Fib4, and NFS in one multi-centre Chinese cohort[73], but needs validation in other populations.

It must be recognised that many scores for assessment of fibrosis and/or steatosis may be specifically influenced by CHB due to their reliance on laboratory parameters which can be independently affected by HBV infection (e.g., liver enzymes and/or platelets). Other conditions that impact platelet count should also be considered, which vary widely by clinical population. Score performance can also potentially be negatively impacted as the number of cardiometabolic risk factors increases; this has been described for Fib4, APRI, and NFS[54].

## New technologies to diagnose MASLD

A machine learning approach, the 'Gradient Boosting Classifier' (GBC), has been trialled to predict MASH in a Chinese cohort of almost 2000 treatment-naïve PLWHB and biopsy-proven MASLD. This approach demonstrated a high AUC and negative predictive values across a training and two validation cohorts (Table 4)[74]. While these results are promising, vast data and computing requirements are likely to limit global application of the technique.

Other developing technologies, for example, metabolomics, can be applied to MASLD diagnosis[75]. The metabolomics-advanced steatohepatitis fibrosis score (MASEF) uses 12 lipids, together with BMI, AST and ALT to predict at-risk MASH (NAFLD Activity Score ≥4 and $F \geq 2$ fibrosis) with an AUC 0.79 (95% CI 0.75–0.83), sensitivity 78%, specificity 65%, positive predictive value 48% and negative predictive value 73% in an international validation cohort[76]. However, this study did not assess MASEF performance in PLWHB. Similar to machine learning approaches, widespread roll-out of this technology is currently limited by the requirement for high-cost specialist equipment and expertise for data analysis, but the landscape is changing rapidly with a focus on delivery of more personalised risk stratification and management, supported by the potential for AI—approaches to analysis and interpretation of large '-omics' data sets.

## Refining diagnostic approach to the context and setting

The context in which clinical assessment takes place currently informs the choice of assessment strategy, for example, the extent to which imaging is available and affordable, both at individual and population levels in community settings, primary care, or through specialist services. MRI- and CT-based imaging techniques are the most accurate non-invasive methods to diagnose steatosis and fibrosis, with some arguing that MRI-PDFF should be the new gold standard for steatosis clinical trial endpoints[77]. However, most clinicians providing care for PLWHB will not have easy access to these imaging modalities, which are high-cost, available only in specialist centres, and reserved for specialist hepatology assessment.

**Table 4 | Serum-based non-invasive tests to diagnose fibrosis in people living with metabolic dysfunction-associated steatotic liver disease (MASLD) and chronic hepatitis B (CHB)**

| Test | Components | Data for CHB + MASLD | Comments |
|---|---|---|---|
| Fibrosis-4 (Fib-4) | AST, ALT, platelets, Age | 8 studies (total n = 5799)[53,54,62–64,66,73,81]. AUC range: 0.65–0.78, Sensitivity range: 56–90%, Specificity range: 36–90%. Evidence for decreased AUC and specificity as the number of cardiometabolic factors increases[54]. | Externally validated to detect advanced fibrosis ($F \geq 3$) for different disease aetiologies. Recommended in EASL guidelines to risk-stratify MASLD in primary care[47] Decreased diagnostic performance <35 and >65 years[45]. |
| Aspartate aminotransferase to platelet ratio index (APRI) | AST, platelets | 7 studies (total n = 5683)[54,62–64,66,73,81] (14–20). AUC range: 0.60–0.77, Sensitivity range: 58–69%, specificity range: 59–80%. Evidence for decreased AUC and specificity as the number of cardiometabolic factors increases[54]. | Recommended by WHO to assess liver fibrosis in CHB treatment guidelines[2] |
| Gamma-glutamyl transferase to platelet ratio (GPR) | GGT, platelets | 3 studies (total n = 3222)[53,64,66]. AUC range: 0.77–0.89, Sensitivity range 66–83%/specificity range 63–80%[66]. Similar performance to Fib-4 and APRI overall (but better performance than VCTE and Fib4 in MetALD) in a large Chinese cohort[64]. | Poor performance in HIV/HBV coinfection[69]. Less widely validated than APRI and Fib-4. Main application within research. |
| NAFLD fibrosis score (NFS) | AST to ALT ratio, platelets, albumin, blood glucose, age, and BMI. | 4 studies (total n = 3676)[53,54,62,73]. AUC range: 0.57–0.72. Evidence for decreased AUC and specificity as the number of cardiometabolic factors increases[54]. | Reduced diagnostic performance <35 and >65 years. Widely validated across multi-ethnic MASLD cohorts[45]. |
| PPDHG score | Platelets, prothrombin time, diabetes, HBeAg, immunoglobulin | One study (n = 504)[73]. AUC: 0.82 (95% CI 0.75–0.89), Sensitivity: 71%, Specificity: 76%. Performed better than NFS, Fib-4, and APRI in one cohort. | Specifically developed to detect advanced fibrosis in antiviral naïve CHB + MASLD. Needs validation in other studies. |
| Gradient boosting classifier (GBC) score | AST, platelets, prothrombin time, albumin, HBeAg status, HBsAg, WBC, INR, BMI, and total bilirubin. | One study (n = 1787)[74]. AUC 0.76–0.89, Sensitivity 58–85%, specificity 79–81% (range across training and validation cohorts). | Machine learning-based model to predict moderate-severe inflammation in antiviral naïve CHB + MASLD (i.e., MASH). Freely available online tool to calculate the score. Needs external validation. |
| AGILE3+ | VCTE, AST, ALT, platelets, diabetes, age, sex | One study[63]. AUC 0.83 (95% CI 0.80–0.87). Similar AUC to VCTE alone (0.83 vs. 0.80). Fewer indeterminate cases (33% vs. 42% VCTE alone). | Still requires Fibroscan, so more expensive than serum-based tools. |
| Enhanced liver fibrosis (ELF) | Quantitative measurements of 3 serum markers of collagen metabolism (HA, PIIINP, TIMP-1) | None in CHB/MASLD. For advanced fibrosis: CHB alone AUC 0.74 (0.63–0.81); MASLD alone AUC 0.80 (0.73–86)[101] | Recommended as an alternative to VCTE in EASL MASLD guidelines. Not routinely measured serum markers, therefore not widely available, especially in resource-constrained settings. |

*ALT* alanine transaminase, *APRI* aspartate aminotransferase-to-platelet-ratio, *AST* aspartate aminotransferase, *AUC* area under the receiver operator curve analysis, *BMI* body mass index, *CHB* chronic hepatitis B, *EASL* European Association for the Study of the Liver, *ELF* enhanced liver fibrosis, *Fib-4* Fibrosis-4 score, *GBC* gradient boosting classifier, *GGT* gamma-glutamyl transferase, *GPR* gamma-glutamyl transferase to platelet ratio, *HA* hyaluronic acid, *HBeAg* hepatitis B e antigen, *HBsAg* hepatitis B surface antigen, *HBeAg* hepatitis B e antigen, *INR* international normalised ratio, *MASH* metabolic dysfunction-associated steatohepatitis, *MASLD* metabolic dysfunction-associated steatotic liver disease, *NAFLD* non-alcoholic fatty liver disease, *NFS* NAFLD fibrosis score, *PIIINP* amino-terminal propeptide of type III procollagen, *TIMP-1* tissue inhibitor of metalloproteinase 1, *VCTE* vibration controlled transient elastography, *WBC* white blood cells.

Scores based on routinely collected clinical information are pragmatic and easy to use; however, they are limited by low specificity[45]. American and European guidelines suggest using serum-based NITs to screen for fibrosis in individuals at high-risk for MASLD in primary care[47,78]. To address the issue of diagnostic accuracy, two thresholds are adopted: a lower threshold with high specificity to rule-out fibrosis, and a higher threshold with greater sensitivity to rule-in fibrosis and qualify for immediate hepatology referral[47]. This leaves an "intermediate zone" of individuals, who should undergo a further NIT to risk-stratify them for hepatology referral or increased surveillance in primary care.

Although MASLD screening in the general population is not recommended[47,78], there is an argument to screen for MASLD among PLWHB. Although only some studies suggest worse liver outcomes with concomitant MASLD and CHB, the 2024 WHO CHB treatment guidelines suggest MASLD as a potential antiviral criterion[2]. Extra-hepatic MASLD manifestations pose a significant health threat (e.g., increased risk of extra-hepatic cancer and cardiovascular events)[5], and knowledge of a MASLD diagnosis presents an opportunity to optimise management of comorbidities and may help clinicians interpret persistently abnormal liver function tests. Moreover, ultrasound and Fibroscan™ assessment are routine components of CHB monitoring in high-income settings; therefore, MASLD diagnosis will only require additional screening for cardiometabolic risk factors.

Access to ultrasound-based diagnostics is severely limited in low-income settings, and in this context, serum-based scores to screen for steatosis may be beneficial. Although specific tests to diagnose MASLD in PLWHB (e.g., SIHBV) show promising results, they have not been replicated outside a single study. Of all the serum-based NITs available to diagnose steatosis, HSI has the most evidence in CHB and uses commonly measured parameters; therefore, it may be a pragmatic suggestion in resource-constrained settings (Table 3, Supplementary Table 3). However, HSI still lacks validation outside of Asia and therefore must be interpreted with caution. Since APRI has been adopted by the WHO in HBV guidelines[2], and has similar performance to Fib-4 in PLWHB/MASLD[53,54,64,66,79–81], it is reasonable to use this to identify individuals at risk of fibrosis.

An example algorithm to screen for MASLD in viral hepatitis clinics, considering both high-income and resource-constrained settings, is suggested in Fig. 5. However, this is not intended to be a guideline for widespread adoption, rather an illustration of how NITs could be implemented based on the limited available evidence. Better evidence for the impact of MASLD on CHB outcomes and the cost-effectiveness of screening is required before implementation.

## A roadmap for the field
### Establishing consistent thresholds and expanding data collection
Validating tests across different populations, and forming expert consensus on consistent approaches will help with understanding the epidemiology

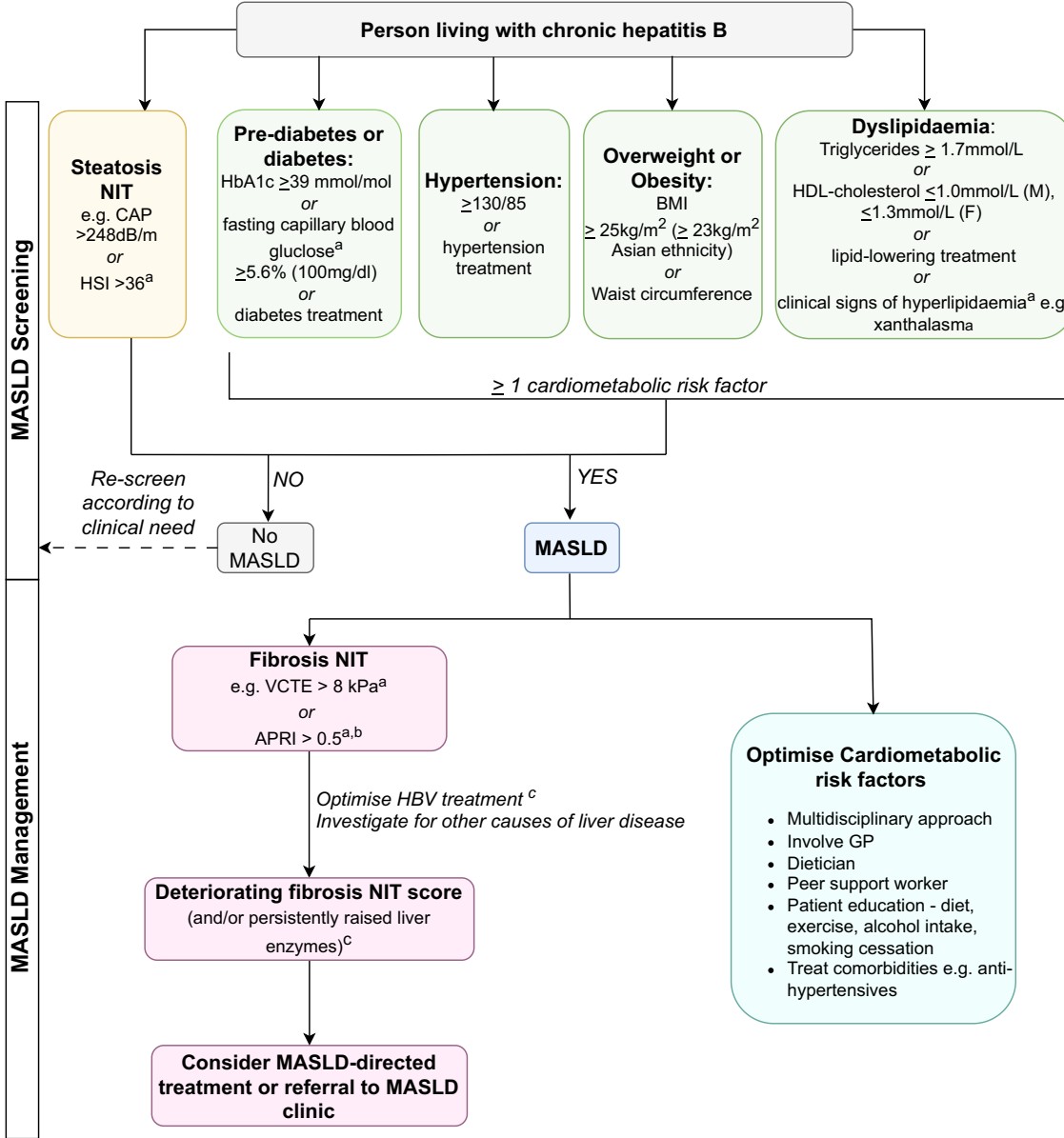

**Fig. 5 | Example algorithm to demonstrate how non-invasive tests (NITs) may be applied to diagnose and risk-stratify metabolic dysfunction-associated steatotic liver disease (MASLD) in people living with chronic hepatitis B (PLWHB).** This is not intended to be a clinical guideline, as further evidence on the impact of MASLD on outcomes in PLWHB and cost-effectiveness analysis is required. Suggested controlled attenuated parameter (CAP) score thresholds are based on a large multi-aetiology analysis containing >1000 PLWHB[52]. The suggested hepatic steatosis index (HSI) threshold is based on a rule-in value from the original study[109]. The suggested aspartate transaminase-to-platelet-ratio (APRI) threshold based on World Health Organization (WHO) chronic hepatitis B (CHB) treatment guidelines[2]. The suggested vibration controlled transient elastography (VCTE) threshold based on European Association for the Study of the Liver (EASL) MASLD guidelines[47]. [a]These tests may be more widely available alternatives in a resource-constrained setting.

[b]Could also consider using Fibrosis-4 (Fib-4) instead of APRI, as they have similar diagnostic performance in PLWHB/MASLD. [c]Consider starting antiviral treatment. [d]Persistently elevated liver enzymes: alanine transaminase (ALT), gamma-glutamyl transferase (GGT), or aspartate transaminase (AST), e.g., above the upper limit of normal twice in 6 months, deteriorating fibrosis score, e.g., increase to >8 kPa or, if known fibrosis, increase >2 kPa since last fibroscan. Figure created in draw.io[110]. APRI aspartate aminotransferase to platelet ratio index, BMI body mass index, CAP controlled attenuation parameter, F female, GP general practitioner, HbA1c glycosylated haemoglobin, HDL high-density lipoprotein, HSI hepatic steatosis index, M male, MASLD metabolic dysfunction-associated steatotic liver disease, NIT non-invasive test, VCTE vibration controlled transient elastography.

and identifying/grading MASLD in PLWHB. More work is needed to develop and validate consistent thresholds to standardise data collection, and evidence is needed to better represent diverse populations. Future studies will be enhanced by wider representation of different geographical settings, children and adolescents, and accounting for varied comorbidities and liver disease aetiologies, HBV genotype, diet/alcohol, host factors (e.g., genetics, immune response), and CHB treatment.

Clinical studies have so far failed to elucidate a clear understanding of the MASLD/CHB relationship, and current in vivo and in vitro models for both CHB and MASLD have limitations to date. However, by harnessing the rapidly developing field of liver experimental models, for example, organoids using primary patient-derived hepatocytes, and precision-cut liver slices, there is increasing potential for more accurate models of disease.

## Approaches to integrated healthcare modelled in other fields

The HIV field sets an example for holistic services that aim to integrate healthcare beyond the assessment and management of chronic viral infection, including long-term interventions to improve metabolic and cardiovascular health. This blueprint could be deployed by CHB services, following the mandate to 'make every contact count' and recognising the disadvantages and barriers some individuals may encounter in accessing consistent health care[12,82]. Basic management of cardiometabolic factors could be implemented through decentralised and multidisciplinary teams, for example, including dieticians, counsellors, and peer support workers. Acknowledging that such implementation comes with resource implications, cost-effectiveness analysis will be required to determine the optimum approaches in different settings, offsetting the fiscal investment in service provision against the costs saved in reducing long-term morbidity and mortality.

## Changes in the therapeutic landscape

One major motivating factor for improved diagnostic assessment of liver disease due to CHB and MASLD is the rapidly transforming landscape of pharmacotherapy for both conditions. CHB management using long-term suppressive nucleos/tide analogue therapy is well established, overlapping with antiretroviral therapy and drawing on drugs with excellent safety profiles and a high genetic barrier to resistance. However, the landscape is fast changing, with the 2024 WHO CHB guidelines expanding and simplifying treatment criteria. For the first time, these include the presence of MASLD as a criterion for initiating antiviral therapy[2], recognising the potential for enhanced risk. On this basis, many more PLWHB will be offered treatment, aligning with global targets for eliminating CHB as a public health threat. The field may change further as there is ongoing research to identify functional cure strategies[83].

New therapies to treat people living with obesity and MASLD are licensed and being rapidly taken up. Semaglutide, a glucagon-like-peptide agonist (GLP-1 agonist), was recommended by the UK's National Institute for Health and Clinical Excellence (NICE) in 2023 to treat people with a BMI ≥ 35 (or ≥30 with a weight-related co-morbidity)[84]. Resmetirom, the first pharmacological treatment for MASLD, was given accelerated approval by the United States Food and Drug Administration in March 2024 for at-risk MASH[85]. Although PLWHB are not currently eligible for this treatment, therapeutic recommendations will continue to change, and therefore, it is crucial to be able to diagnose MASLD accurately and consistently. The field will also be influenced according to the results of clinical trials exploring new drugs and combinations (e.g., dual GLP1/GL agonists, DGAT2i/ACCi combination[86]).

## Recognising global disparities and advancing equitable solutions

There is a critical overlap between global regions with the biggest projected increase in obesity and diabetes, high CHB prevalence, and inequitable access to diagnostic methods and health infrastructure[87,88]. Although steatosis and fibrosis scores are useful and accessible tools, they are outperformed by imaging-based NITs. Therefore, there is a pressing need to address inequitable access to VCTE/CAP measurement. Approaches could include the introduction of subsidies that account for resource constraints, access to loan machines, sharing equipment between services, and/or between clinical and research programmes. New hardware that can measure CAP is emerging on the market, which should promote cost reductions and promise wider access, but may also bring challenges in quality and consistency.

In the light of the sudden, drastic withdrawal of US federal funding for overseas clinical and research programmes during 2025, coupled with significant cuts to foreign aid from several European countries, existing global health disparities are set to deepen[89]. This underscores the urgent need for healthcare and scientific communities to speak with a unified voice to retain global health funding and international research collaboration[90]. Improved diagnosis and research representation across diverse global settings will clarify the scope of the problem and support calls for expanding funding and investment needed to address the growing challenge posed by concomitant MASLD and CHB.

## Conclusions

Many tools are available for clinical assessment of liver disease, but there is a pressing need to advance the field by unifying classification systems, expanding access, and collecting consistent data. These are urgent public health imperatives, given the increasing morbidity and mortality associated with both CHB and metabolic disease. There are substantial knowledge gaps regarding the mechanisms and impact of the interaction between these conditions in individuals and at a population level. As new tests, risk-stratification tools and algorithms emerge, their efficacy should be measured not only by their predictive accuracy in liver disease but also by their global accessibility and potential value in evaluating the impact of viral hepatitis infection on metabolic and cardiovascular healthcare.

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

## Acknowledgements

P.C.M. receives funding from the Francis Crick Institute which receives its core funding from Cancer Research UK, the UK Medical Research Council and the Wellcome Trust (ref. CC2223) and is also supported by University College London NIHR Biomedical Research Centre. E.M. is a doctoral clinical fellow funded by the Francis Crick Institute.

## Author contribution

The article was conceived by E.M. and P.C.M. The primary literature review and manuscript draft were undertaken by E.M., supervised by P.C.M. Figures were made by E.M. and P.C.M. A.A.P., R.G., N.C., S.F., D.M., E.A.T., W.D.F.V., and J.M.G. contributed to refining the text, citations, and presentation of the final article.

## Competing interests

P.C.M. has received funding support from GSK for a member of her team (2019-2023) outside the scope of this paper and has received payment from J&J for delivery of educational material during 2025. P.C.M. is clinical co-lead for the NIHR Health Informatics Collaborative for viral hepatitis and liver disease which has received GSK funding support, and is co-chair of the National Strategic Group for Viral Hepatitis. A.A.P. reports investigator-initiated study grants and/or personal fees from ViiV Healthcare, Gilead Sciences, and Jansen Pharmaceuticals (now Johnson & Johnson Innovative Medicine), outside the scope of this paper. W.D.F.V.'s unit receives funding from the Bill and Melinda Gates Foundation, SA Medical Research Council, National Institutes for Health, Unitaid, Foundation for Innovative New Diagnostics (FIND), Merck and the Children's Investment Fund Foundation (CIFF), has previously received funding from USAID, and received drug donations from ViiV Healthcare, Merck, J&J and Gilead Sciences for investigator-led clinical studies. This unit does investigator-led studies with Merck, J&J, Gilead, and ViiV, providing financial support, and is doing commercial drug studies for Merck and Novo, and performs evaluations of diagnostic devices for multiple biotech companies. Individually, W.D.F.V. receives honoraria for educational talks and advisory board membership for Gilead, ViiV, Mylan/Viatris, Merck, Adcock-Ingram, Aspen, Abbott, Roche, J&J, Sanofi, Boehringer Ingelheim, Thermo-Fischer, and Virology Education. J.M.-G. is a site Principal Investigator for clinical trials with Viiv, GSK, Gilead, and Merck that fall outside the scope of this paper. N.C.'s unit receives research funding from Gilead and has received honoraria from Gilead Sciences and Novo Nordisk. R.G.'s unit receives research funding from Gilead, GSK, ViiV Healthcare, and Janssen. S.F. has received honoraria, conference support, and research grants from Viiv, Gilead, and MSD. Other authors have no competing interests to declare.
