## [Transparent Peer Review file · Communications Medicine]

Diagnosing and defining MASLD in people living with chronic hepatitis B

Corresponding Author: Professor Philippa Matthews

Version 0:

Reviewer comments:

Reviewer #1

(Remarks to the Author)

This narrative review addresses an increasingly important intersection in hepatology: the diagnosis and risk stratification of MASLD in individuals with CHB. The authors provide a well-structured overview of non-invasive tools and thoughtfully address health disparities, evolving nomenclature, and the limitations of current approaches. I hope the authors may find some of my suggestions useful.

1. One major conceptual insight currently underemphasized in the manuscript is the diverging impact of hepatic steatosis and metabolic dysfunction on HBV-related outcomes. While hepatic steatosis appears to promote HBsAg seroclearance and may even reduce the risk of cirrhosis, HCC, and mortality in patients with CHB (a pattern that contrasts with its adverse role in the general population), metabolic dysfunction (particularly DM) has been consistently associated with worse outcomes in CHB. These opposing effects represent a central paradox in the CHB-MASLD literature and are likely responsible for the conflicting results seen across existing studies. A key determinant of this heterogeneity is whether the original studies analyzed hepatic steatosis and metabolic dysfunction as distinct variables, or instead conflated them under the umbrella of "MASLD" or prior NAFLD definitions. Studies that did not distinguish these two biological processes may report mixed conclusions. The authors are encouraged to make this distinction explicit in the manuscript and discuss the important papers in this field. Furthermore, this distinction has important implications for clinical risk stratification and therapeutic decision-making.

References (recent studies that distinguished steatosis and metabolic dysfunction for analyses, showing the opposite effects of the two factors in CHB): PMID 39675434, PMID 40091278, PMID 38920306

2. The manuscript provides a comprehensive summary of imaging and serum-based NITs, such as FibroScan, MRI-PDFF, HSI, FLI, NAFLD-LFS, SIHBV, and others. To enhance the review's practical value, it would be helpful if the authors could offer more expert interpretation regarding the clinical utility of each test (for example, which tools are best suited for primary care settings or certain conditions, which require specialist-level interpretation, and in what sequence or combinations they may be most appropriately used) for hepatic steatosis and fibrosis evaluation.

3. Similarly, practical clinical algorithms are needed to guide clinicians in resource-stratified settings (e.g., how to use Fib-4/APRI in primary care and when to escalate to imaging or biopsy).

Reviewer #3

(Remarks to the Author)

Nice, timely and concise narrative review summarizing the use of NITs to identify steatosis and liver fibrosis in people with concomitant MASLD and HBV. Below are my comments:

1. Given the large number of non-invasive tests available, the article will be enhanced by adding in the perspectives of the authors regarding their recommendations on which to be used if available.
2. "NITs can assess the whole liver" – this is only partly correct – for MRI studies yes as you can choose the region of interest, but not for VCTE because it is just assessing that single point of contact between the probe and the liver surface underneath. Also, VCTE has other limitations e.g., high ALT, ascites, cholestasis, failure rates esp in obese patients. These should also be discussed in the relevant paragraphs.

3. The contradicting results in the outcome of HCC in those with concomitant HBV and MASLD should be further elaborated. Is it related to the different MASLD diagnostic methods? Patient population heterogeneity?

4. VCTE cut-offs: the authors emphasized the potential influence of concomitant MASLD on accuracy of VCTE parameters to assess fibrosis and steatosis in HBV infection, E.g., HBV independently lowers CAP scores. In my humble opinion, as the HBV field is moving towards simplification of assessment and treatment algorithms, establishing a new cut-off for MASLD+HBV will inevitably introduce more confusion and inconsistency among physicians. Perhaps it will be better to simplify things and to stick with the existing criterion for HBV for those with also concomitant MASLD.

Reviewer #4

(Remarks to the Author)

This thorough and well-organised review tackles a significant and developing area of hepatology: the confluence of MASLD and CHB. The authors offer a comprehensive overview of the literature, point out important issues, and suggest a course for further study and clinical application. The manuscript has clear writing and is well-referenced. I have some comments

- Introduction (Lines 44-103) The discussion on heterogeneity in diagnostic tools could be expanded—why do different regions use different thresholds?
- The lack of representation from Africa/Middle East is critical—could this be linked to funding disparities or infrastructure limitations?
- Mechanisms (Lines 126-129):
- The proposed mechanisms for HBsAg loss (e.g., fatty acid inhibition) are intriguing but speculative—could this be framed more cautiously?
- CAP thresholds (Lines 183-201): The discussion on HBV lowering CAP scores is important but could be expanded—does this mean MASLD is underdiagnosed in CHB?
- Fibroscan limitations (Lines 166-169): The cost/access issue is critical—could this be tied to global health equity more explicitly?
- GPR vs. Fib4/APRI (Lines 242-247): The claim that GPR performs better needs more supporting evidence—is this consistent across populations?
- Machine learning (Lines 267-271): The Gradient Boosting Classifier is interesting, but how feasible is this in low-resource settings?
- MASEF score (Lines 282-285): Since this has not been tested in CHB, should this be mentioned as a limitation?

Version 1:

Reviewer comments:

Reviewer #1

(Remarks to the Author)

Thanks a lot for your responsive revision. I have no more comments.

Reviewer #3

(Remarks to the Author)

The revised manuscript reads well. The authors have addressed all comments.

Reviewer #4

(Remarks to the Author)

Dear Editorial Team and Reviewers,

REF: Diagnosing and defining MASLD in chronic hepatitis B: a narrative review of non-invasive tests (COMMSMED-25-0645)

Thank you for taking the time to read and provide thoughtful comments on our paper entitled “Diagnosing and defining MASLD in chronic hepatitis B: a narrative review of non-invasive tests”. Please see below for the point-by-point response. Our responses and explanation are in red text, and quotes from the manuscript are in italic blue text. Edited text the manuscript is highlighted in yellow.

Reviewer #1 (Remarks to the Author):

This narrative review addresses an increasingly important intersection in hepatology: the diagnosis and risk stratification of MASLD in individuals with CHB. The authors provide a well-structured overview of non-invasive tools and thoughtfully address health disparities, evolving nomenclature, and the limitations of current approaches. I hope the authors may find some of my suggestions useful.

1. One major conceptual insight currently underemphasized in the manuscript is the diverging impact of hepatic steatosis and metabolic dysfunction on HBV-related outcomes. While hepatic steatosis appears to promote HBsAg seroclearance and may even reduce the risk of cirrhosis, HCC, and mortality in patients with CHB (a pattern that contrasts with its adverse role in the general population), metabolic dysfunction (particularly DM) has been consistently associated with worse outcomes in CHB. These opposing effects represent a central paradox in the CHB-MASLD literature and are likely responsible for the conflicting results seen across existing studies. A key determinant of this heterogeneity is whether the original studies analyzed hepatic steatosis and metabolic dysfunction as distinct variables, or instead conflated them under the umbrella of “MASLD” or prior NAFLD definitions. Studies that did not distinguish these two biological processes may report mixed conclusions. The authors are encouraged to make this distinction explicit in the manuscript and discuss the important papers in this field. Furthermore, this distinction has important implications for clinical risk stratification and therapeutic decision-making. References (recent studies that distinguished steatosis and metabolic dysfunction for analyses, showing the opposite effects of the two factors in CHB): PMID 39675434, PMID 40091278, PMID 38920306

Thank you for highlighting this perspective and providing references for recent papers. We added a sentence in section 2.1 to explain this observation and added the suggested references. We have also added the point that inferring causation from these complex clinical data (which are often not routinely collected or are collected using heterogeneous approaches) may be problematic and also contributes to ambiguity in the field:

Another possible explanation is that risk factors have a differential impact on outcomes in CHB, with cardiometabolic factors having a dose-dependent adverse effect, while steatotic liver disease may have a protective effect²⁵⁻²⁷. The challenge of accurately estimating the individual causal effect estimates of these interrelated factors using routinely collected clinical data likely contributes to current ambiguity in the literature.

2. The manuscript provides a comprehensive summary of imaging and serum-based NITs, such as FibroScan, MRI-PDFF, HSI, FLI, NAFLD-LFS, SIHBV, and others. To enhance the review's practical value, it would be helpful if the authors could offer more expert interpretation regarding the clinical utility of each test (for example, which tools are best suited for primary care settings or certain conditions, which require specialist-level interpretation, and in what sequence or combinations they may be most appropriately used) for hepatic steatosis and fibrosis evaluation.

Please see the response to point 3, as these suggestions are tackled together.

3. Similarly, practical clinical algorithms are needed to guide clinicians in resource-stratified settings (e.g., how to use Fib-4/APRI in primary care and when to escalate to imaging or biopsy).

We have tackled point 2 and 3 together. The heading "Refining approach to the context and setting" has been moved from section 4 "future directions" to section 3 "assessment of MASLD in PLWHB", enabling more detailed discussion. We now provide an overview of how these tests are currently used in practice (i.e. primary versus specialist hepatology care), and examples of how they could be implemented in PLWHB in both low- and high-income settings, based on the limited available evidence. For clarity, we added a new figure (Figure 5), which includes an example algorithm of how NITs could be used in viral hepatitis clinics to diagnose and risk stratify MASLD in PLWHB.

We have thought carefully about adding an algorithm, since evidence is limited and we need to avoid this being interpreted as a new 'guideline' for clinical practice. However, we agree it is useful for readers to appreciate how NITs can be applied in clinical practice.

Therefore, we now include an example algorithm, but with a caveat in the text and figure legend that this is not intended for widespread adoption without further evidence base (e.g. cost analysis, clearer understanding of the clinical impact of having both liver diseases etc.)

3.5 Refining approach to the context and setting

The context in which clinical assessment takes place currently informs the choice of assessment strategy, for example, the extent to which imaging is available and affordable, both at individual and population levels in community settings, primary care, or through specialist services. MRI- and CT-based imaging techniques are the most accurate non-invasive methods to diagnose steatosis and fibrosis, with some arguing that MRI-PDFF should be the new gold standard for steatosis clinical trial endpoints⁷⁷. However, most clinicians managing PLWHB will not have easy access to these imaging modalities which are high cost, available only in specialist centres, and reserved for specialist hepatology assessment.

Scores based on routinely collected clinical information are pragmatic and easy to use, however, they are limited by low specificity⁴⁵. American and European guidelines suggest using serum-based NITs to screen for fibrosis individuals at high-risk for MASLD in primary care^{47,78}. To address the issue of diagnostic accuracy, two thresholds are adopted: a lower threshold with high specificity to rule-out fibrosis, and a higher threshold with greater sensitivity to rule-in fibrosis and qualify for immediate hepatology referral⁴⁷. This leaves an "intermediate zone" of individuals, who should

undergo a further NIT to risk stratify them for hepatology referral or increased surveillance in primary care.

Although MASLD screening in the general population is not recommended^{47,78}, there is an argument to screen for MASLD among PLWHB. Although only some studies suggest worse liver outcomes with concomitant MASLD and HBV, 2024 WHO treatment guidelines suggest MASLD as a potential antiviral criterion². The evidence for extra-hepatic MASLD manifestations is strong (e.g. increased risk of extra-hepatic cancer and cardiovascular events)⁵ and knowledge of a MASLD diagnosis presents an opportunity to optimise management of comorbidities and may help clinicians interpret persistently abnormal liver function tests. Moreover, ultrasound and FibroscanTM assessment are often routine components of CHB monitoring in high-income settings, therefore MASLD diagnosis will only require additional screening for cardiometabolic risk factors.

*Access to ultrasound-based diagnostics is severely limited in low-income settings, and in this context serum-based scores to screen for steatosis may be beneficial. Although specific tests to diagnose MASLD in PLWHB (e.g. SIHBV) show promising results, they have not been replicated outside a single study. Of all the serum-based NITs available to diagnose steatosis, HSI has the most evidence in CHB and uses commonly measured parameters, therefore may be a pragmatic suggestion in resource-constrained settings (**Table 3A, Suppl Table 3**). However, HSI still lacks validation outside of Asia and therefore must be interpreted with caution. Since APRI has been adopted by the WHO in HBV guidelines², and has similar performance to Fib-4 in PLWHB/MASLD^{53,54,64,66,79–81}, it is reasonable to use this to identify individuals at risk of fibrosis.*

*An example algorithm to screen for MASLD in viral hepatitis clinics, considering both high-income and resource-constrained settings, is suggested in **Figure 5**. However, this is not intended to be a guideline for widespread adoption, rather an illustration of how NITs could be implemented based on the limited available evidence. Better evidence for the impact of MASLD on HBV outcomes and cost-effectiveness of screening is required before implementation.'*

Reviewer #2 (Remarks to the Author):

Nice, timely and concise narrative review summarizing the use of NITs to identify steatosis and liver fibrosis in people with concomitant MASLD and HBV. Below are my comments:

1. Given the large number of non-invasive tests available, the article will be enhanced by adding in the perspectives of the authors regarding their recommendations on which to be used if available.

Thank you for this comment, which is in line with comments we have received from other reviewers (see discussion with reviewer 1, above). We agree that an illustration of how these tests are used will add to the overall practical value of the paper and have therefore added a new section (Section 3.5: Refining approach to the context and setting) and new **Figure 5**, with an example algorithm of how NITs could be implemented in viral hepatitis clinic to diagnose and risk stratify MASLD in PLWHB.

2. 'NITs can assess the whole liver' – this is only partly correct – for MRI studies yes as you can choose the region of interest, but not for VCTE because it is just assessing that single point of contact between the probe and the liver surface underneath. Also, VCTE has other limitations e.g., high ALT, ascites, cholestasis, failure rates esp in obese patients. These should also be discussed in the relevant paragraphs.

We have deleted the sentence "In contrast to biopsy, NITs can assess the whole liver" in section 3.1. These important limitations of VCTE have been added into the text in paragraph two, section 3.2 as follows:

Other limitations include reduced accuracy caused by factors which affect the elastic properties of the liver (e.g. transaminitis, congestive heart failure, post-prandial hyperaemia, cholestasis) and impede the transmission of shear waves (e.g. ascites)⁴¹. The standard M probe may not perform well in the context of overweight/obesity, therefore Fibroscan™ introduced the XL probe for use when skin-to-capsule distance is >25mm⁴².

We have also amended **Table 2** accordingly.

3. The contradicting results in the outcome of HCC in those with concomitant HBV and MASLD should be further elaborated. Is it related to the different MASLD diagnostic methods? Patient population heterogeneity?

This is a large topic and the subject of much debate, therefore critically evaluating all the possible reasons in detail is beyond the scope of this review (another detailed review is needed to address this question in full). However, we have added more detail to section 2.1 to highlight some possible explanations e.g. heterogeneity of studies, evidence for differential outcomes of steatosis and cardiometabolic risk factors, difficulty in accurately analysing routinely collected data in this complex field. Additional sentences from section 2.1 are as follows:

*The reasons behind these conflicting findings are likely to be multifactorial. One possible contributing factor is the heterogeneity of studies included in the meta-analyses. For example, older definitions of metabolic or fatty liver disease (**Figure 2**), different methods of MASLD diagnosis (e.g. imaging, biopsy, non-invasive serum-based scores), different populations, and varied study designs (e.g. cohort vs cross sectional) (**Table 1**).*

4. VCTE cut-offs: the authors emphasized the potential influence of concomitant MASLD on accuracy of VCTE parameters to assess fibrosis and steatosis in HBV infection, E.g., HBV independently lowers CAP scores. In my humble opinion, as the HBV field is moving towards simplification of assessment and treatment algorithms, establishing a new cut-off for MASLD+HBV will inevitably introduce more confusion and inconsistency among physicians. Perhaps it will be better to simplify things and to stick with the existing criterion for HBV for those with also concomitant MASLD.

We agree that adding further stratification to an already complex field, in the context of global efforts to simplify and widen access to HBV treatment, may not be helpful to readers. In fact, we already argue for consistent CAP thresholds to improve data collection and clinical management in section 4.1:

More work is needed to develop and validate consistent thresholds to standardise data collection, and evidence is needed to better represent diverse populations.

We still believe it is important to present the available evidence regarding thresholds in CAP and VCTE. We agree that multiple thresholds create additional confusion, but there is currently not enough evidence to suggest a single threshold that applies across all liver diseases. To address this important point, we added the following statement in section 3.3:

While data to suggest variation in CAP and VCTE values according to disease threshold may create confusion, especially as there are global efforts to simplify HBV treatment guidelines, it is not possible to suggest a single cut off threshold based on our current data. Further research is needed to understand optimal thresholds for use in clinical practice.

To maintain consistency with EASL MASLD treatment guidelines, we have chosen 8kPa as a cut-off for MASLD-directed management/ MASLD clinic referral in our example algorithm (**Figure 5**), noting however, that WHO guidelines use 7kPa, and that further research is needed.

Reviewer #3 (Remarks to the Author):

This thorough and well-organised review tackles a significant and developing area of hepatology: the confluence of MASLD and CHB. The authors offer a comprehensive overview of the literature, point out important issues, and suggest a course for further study and clinical application. The manuscript has clear writing and is well-referenced. I have some comments

- Introduction (Lines 44-103)The discussion on heterogeneity in diagnostic tools could be expanded—why do different regions use different thresholds?

We have tackled this point in more detail Section 3.3 “Imaging thresholds for fibrosis and steatosis”. We explain that each guideline has chosen different studies upon which to base their recommendation and have explained some of the differences between the studies. We also explain that the American guidelines use a rule-in threshold, while the European guidelines define thresholds best on steatosis severity. These thresholds are also visualised in Figure 4. We have reworded section 3.3. paragraph 1 to improve clarity, as follows:

*Clinical and public health applications of CAP are currently limited by lack of consistency for CAP thresholds in comparison to assessment by biopsy (**Figure 4**). European MASLD guidelines suggest distinct thresholds for distinguishing between steatosis grades (with cut-offs to distinguish S0/1, S1/2 and S2/3 thresholds), based on individual patient data meta-analysis including almost 4000 individuals (with multi-aetiology liver disease including >1000 people living with HBV). In contrast, American guidelines advocate for a single rule-in CAP threshold which sits in the European S3 category (based on a single centre study with 119 participants, excluding PLWHB)^{46–49}. The decision to select a single binary threshold recognises the limited accuracy of CAP to distinguish between steatosis grades, in particular S2 to S3^{50,51}.*

- The lack of representation from Africa/Middle East is critical—could this be linked to funding disparities or infrastructure limitations?

This is a critical point, and we make multiple references to disparity in access to diagnostic equipment and lack of representation throughout the original manuscript. However, in response to this comment, we have consolidated several of these points into a new section 4.4 “recognising global disparities and advancing equitable solutions”. In this section, we emphasise that the overlap between MASLD and CHB in low-resource settings, highlight the importance of equity in access to diagnostic tools and infrastructure and describe how improved diagnostics and representation in research is essential to recognise the scope of the problem and direct funds and investment appropriately. We have also mentioned the negative impact the current political landscape is having on global health funding and opportunities international collaboration, advocating for the scientific community to speak out against these changes. This section reads as follows:

4.4 Recognising global disparities and advancing equitable solutions

There is a critical overlap between global regions with the sharpest projected increase in obesity and diabetes, high HBV prevalence, and inequitable access to diagnostic methods and health infrastructure^{87,88}. Although steatosis and fibrosis scores are useful and accessible tools, they are outperformed by imaging-based NITs. Therefore, there is a pressing need to address inequitable access to VCTE/CAP measurement. Approaches could include the introduction of subsidies that account for resource constraints, access to loan machines, sharing equipment between services and/or between clinical and research programmes. New hardware that can measure CAP is emerging onto the market, which should promote cost reductions and promise wider access, but may also bring challenges in quality and consistency.

In the light of the sudden, drastic withdrawal of US federal funding for overseas clinical and research programmes, coupled with significant cuts to foreign aid from several European countries, existing global health disparities are set to deepen⁸⁹. This underscores the urgent need for healthcare and scientific communities to speak with a unified voice in defence of global health funding and international research collaboration⁹⁰. Improved diagnosis and research representation across diverse global settings will clarify the scope of the problem and support calls for expanding funding and investment needed to address the growing challenge posed by concomitant MASLD and HBV.

- Mechanisms (Lines 126-129):
- The proposed mechanisms for HBsAg loss (e.g., fatty acid inhibition) are intriguing but speculative—could this be framed more cautiously?

We have rephrased section 2.1 to reflect the speculative nature of this observation, as follows:

One consistent (albeit perhaps unexpected) observation is that clearance of hepatitis B surface antigen (HBsAg) is more likely in the presence of MASLD^{17,19}. Hypothesised mechanisms for this increased chance of HBsAg loss (also termed ‘functional cure’) include increased apoptosis of steatotic HBV-infected hepatocytes, fatty acid inhibition of HBsAg secretion, and increased MASLD-related inflammation boosting immune-related HBsAg seroclearance^{28–31}. However, these mechanisms are speculative and more robust experimental evidence is required.

- CAP thresholds (Lines 183-201): The discussion on HBV lowering CAP scores is important but could be expanded—does this mean MASLD is underdiagnosed in CHB?

It is possible that higher CAP thresholds may lead to underdiagnosis of mild steatosis in PLWHB, however the clinical importance of this is currently unclear. We have acknowledged that the literature is confusing, and suggest that further research is needed:

While data to suggest variation in CAP and VCTE values according to disease threshold may create confusion, especially as there are global efforts to simplify HBV treatment guidelines, it is not possible to suggest a single cut off threshold based on our current data. Further research is needed to understand optimal thresholds for use in clinical practice.

- Fibroscan limitations (Lines 166-169): The cost/access issue is critical—could this be tied to global health equity more explicitly?

In line with the previous point on funding disparities and infrastructure, this point has been specifically addressed under the new section 4.2 “recognising global disparities and advancing equitable solutions” (see manuscript quote above).

- GPR vs. Fib4/APRI (Lines 242-247): The claim that GPR performs better needs more supporting evidence—is this consistent across populations?

In section 3.4 (ii) “scores to identify fibrosis” paragraph 2, we do already provide details of 4 studies that report improved performance of GPR compared to Fib-4 and APRI in individual studies and certain populations (e.g. PLWHB alone and PLWHB/MetALD):

In some datasets, GPR may perform better than other laboratory-based approaches in identifying advanced fibrosis (F3-F4)⁶⁵. In a small Chinese study, GPR had a significantly higher AUC and negative predictive value to detect $\geq F3$ fibrosis compared to Fib-4 and APRI (Table 3B, Suppl Table 4)⁶⁶, and in African cohorts GPR has also outperformed other scores^{67,68}.

Further details of the studies can be found in **Table 3B** and **Supplementary Table 4** We have now also added a sentence to clarify that this test is understudied compared to other fibrosis NITs and is not suitable for HIV/HBV coinfection.

However, GPR remains understudied compared to other NITs and has poor performance in the setting of HBV/HIV coinfection, limiting widespread implementation⁶⁹.

- Machine learning (Lines 267-271): The Gradient Boosting Classifier is interesting, but how feasible is this in low-resource settings?

For completeness, we have included experimental tests using newer technology. However, to address this point we have added a sentence about feasibility, highlighting that this approach is interesting, but will likely not be applicable in resource-constrained settings.

While these results are promising, vast data and computing requirements are likely to limit global application of the technique.

- MASEF score (Lines 282-285): Since this has not been tested in CHB, should this be mentioned as a limitation?

We added a sentence to highlight the limitations of this approach, including that it has not been tested in CHB, and is also unlikely to be available currently for widespread rollout due to high cost:

However, since this study did not assess MASEF performance in CHB, application of the results in PLWHB is limited. Similar to machine learning approaches, widespread roll-out of this technology is currently limited by requirement for high-cost specialist equipment and expertise for data analysis, but the landscape is changing rapidly with a focus on delivery of more personalised risk stratification and management, supported by the potential for AI-approaches to analysis and interpretation of large ‘-omics data sets.

Since MASEF doesn't have data in CHB/MASLD and is a new technology, we haven't included it in the NIT reference tables.

Other changes

We have made several minor changes not mentioned by the reviewers, listed below:

- Updated the number of systematic reviews as another has been published since submission (section 2.1, table 1, figure 3).
- Added enhanced liver fibrosis (ELF) to the manuscript, table 3B and supplementary table 4. Although there is no evidence in concomitant MASLD/CHB, this test is recommended in European and American MASLD guidelines to detect fibrosis, and has some evidence in CHB and MASLD alone. Therefore we felt this was a relevant minor addition to the paper.
- Recreated Figure 1 in biorender to satisfy the journal's image format requirements.
- Updated author S.F's conflict of interests.

We are grateful to the reviewers for their insightful feedback. Their input has helped to refine and clarify key messages of the review, and we hope the editor will consider the manuscript favourably for publication.

Yours sincerely,

Emily Martyn

On behalf of Alejandro Arenas-Pinto, Richard Gilson, Nomathemba Chandiwana, Stuart Flanagan, Douglas MacDonald, Emmanuel A. Tsochatzis, W. D. Francois Venter, Jennifer Manne Goehler, Philippa C Matthews